# Identification and Testing of Antidermatophytic Oxaborole-6-Benzene Sulphonamide Derivative (OXBS) from *Streptomyces atrovirens* KM192347 Isolated from Soil

**DOI:** 10.3390/antibiotics9040176

**Published:** 2020-04-13

**Authors:** Seham Abdel-Shafi, Abdul-Raouf Al-Mohammadi, Taghreed N. Almanaa, Ahmed H. Moustafa, Tamer M. M. Saad, Abdel-Rahman Ghonemey, Immacolata Anacarso, Gamal Enan, Nashwa El-Gazzar

**Affiliations:** 1Department of Botany and Microbiology, Faculty of Science, Zagazig University, El-Sharqia 44519, Egypt; mora_sola1212@yahoo.com; 2Department of Science, King Khalid Military Academy, Riyadh 11495, Saudi Arabia; almohammadi26@hotmail.com; 3Department of Botany and Microbiology, College of Science, King Saud University, Riyadh 11495, Saudi Arabia; 4Department of Chemistry, Faculty of Science, Zagazig University, Zagazig 44519, Egypt; ah_hu_mostafa@yahoo.com; 5Nuclear Materials Authority, Cairo 11381, Egypt; 6Department of Life Sciences, University of Modena and Reggio Emilia, Via Campi, 41121 Modena, Italy; immaanacarso@gmail.com

**Keywords:** streptomycetes, *S. atrovirens*, dermatophytes, antifungal activity, OXBS

## Abstract

There is a need to continue research to find out other anti-dermatophytic agents to inhibit causal pathogenic skin diseases including many types of tinea. We undertook the production, purification, and identification of an anti-dermatophytic substance by *Streptomyces atrovirens*. Out of 103 streptomycete isolates tested, only 20 of them showed antidermatophytic activity with variable degrees against *Trichophyton tonsurans* CCASU 56400 (*T. tonsurans*)*, Microsporum canis* CCASU 56402 (*M. canis*), and *Trichophyton mentagrophytes* CCASU 56404 (*T. mentagrophytes*). The most potent isolate, S_10_Q_6_, was identified based on the tests conducted that identified morphological and physiological characteristics and using 16S rRNA gene sequencing. The isolate was found to be closely correlated to previously described species *Streptomyces atrovirens;* it was designated *Streptomyces atrovirens* KM192347 (*S. atrovirens*). Maximum antifungal activity of the strain KM192347 was obtained in modified starch nitrate medium (MSNM) adjusted initially at pH 7.0 and incubated at 30 °C in shaken cultures (150 rpm) for seven days. The antifungal compound was purified by using two steps protocol including solvent extraction and column chromatography. The MIC of it was 20 µg/mL against the dermatophyte cultures tested. According to the data obtained from instrumental analysis and surveying the novel antibiotics database, the antidermatophytic substance produced by the strain KM192347 was characterized as an oxaborole-6-benzene sulphonamide derivative and designated oxaborole-6-benzene sulphonamide (OXBS) with the chemical formula C_13_H_12_ BNO_4_S. The crude OXBS didn’t show any toxicity on living cells. Finally, the results obtained herein described another anti-dermatophytic substance named an OXBS derivative.

## 1. Introduction

Streptomycetes are widely distributed in soil, water, and plants and are Gram-positive, free living saprophytic actinobacteria [1]. They belong to the order Actinomycetales. The Genus *Streptomyces,* has potential ability to produce bioactive compounds with antimicrobial effects and has important applications in human medicine [2]. *Streptomyces* spp. is considered to be the major producers of about half of all naturally occurring antibiotics [3]. *Streptomyces* spp. has recently been shown to produce antibiotics when stressed by alkaliphilic environment [4]. The antifungal activity of *Streptomyces* spp. is mainly related to the production of antifungal compounds and extracellular hydrolytic enzymes [5]. Dermatophytes that have the ability to invade keratinized tissues such as skin, hair and nails of both human and animals cause dermatophytosis. The important dermatophytes genera are *Trichophyton*, *Microsporum*, and *Epidermatophyton* [6]. The detection of antifungal resistance within fungal pathogens is increasing [7]. Therefore, the discovery of new natural antifungal agents with high safety profiles and efficiency against dermatophytesis very important [8].

Until the 1970s, fungal infections were considered largely treatable, and the demand for medicines to treat them was very small. Prior to this period, antifungal chemotherapy included only two kinds of compounds—potassium iodide, which was effective in the treatment of sporotrochosis, and two useful polyenes, nystatin and amphotericin B, which were introduced in the1950s. Except for the development of flucytosine (1964), there was a little progress until the development of azole drugs in the early 1970s [9,10,11,12,13,14]. Therefore, only a limited number of antifungal agents, such as polyenes and azoles, are currently available for the treatment of life-threatening fungal infection. These antifungal agents showed some limitations, such as the significant nephro-toxicity of amphotericin B and emerging resistance to the azoles [5]. Dermatophytes are the caused pathogens of tinea diseases such as tineacapitis, tineapedis, tineacruris, and tineacorporis, which infect the head, foot, public regions, and torso, respectively [15]. Treatment of tinea occurred by topically used azoles and allylamines; oral itraconazole is also used [16]. The treatment of fungal infection of the nail bed (onychomycosis) faces certain challenges because the fungal pathogens colonize the subungual region and cause thickening, discoloration, or cracking of the nail bed, which in turn cause food pain and necrosis around nail bed [15]. Also, the nail bed in cases of onychomycosis makes a barrier for drugs. Due to such challenges and emergence of resistant variantsof microorganisms [17], there is a need to develop novel materials to protect human from microbial infections [18]. Also, the discovered antidermato fungal agents inhibit fungal peptide synthesis [15].

Tavaborole antifungal is a chemically synthesized member of oxaboroles and is used commercially under the name Kerydin by Anacor Pharmaceuticals, Inc. in Palo Alto, California, United States and was first approved by the Food and Drug Administration (FDA) on 7July 2014 [19]; it was found to penetrate through the nail bed and multiple layers of nail polish due to its low molecular weight [20]. Therefore, the discovery of biologically synthesized oxaborole derivative is very promising.

The present study was undertaken to characterize other oxaborole derivative (oxaborole-6-benzene sulphonamide derivative, OXBS), to maximize its production, and to elucidate its structure. The toxicity of OXBS was studied.

## 2. Results

### 2.1. Isolation of Streptomycetes from Soil Samples and Screening Their Antifungal Activity

Almost all the examined soil samples showed positive streptomycete growth. About 103 streptomycete isolates were obtained; their colors of aerial mycelia were either grey, yellow, red, white, blue, or green. These isolates were purified and maintained onto starch nitrate agar slants for further study. Only 20 isolates (19.4%) showed antidermatophytic activity ofvariable degrees against the three indicator dermatophytes tested (Table 1). Results have showed that the highest antifungal activity was observed from the culture of isolate S_10_Q_6_ (*p* value ≤ 0.01) (Table 1). This isolate was chosen for further experimental studies.

### 2.2. Characterization and Identification of the Most Active Streptomycetes Isolate

The isolate S_10_Q_6_was Gram positive, and its aerial hyphae bear oval spore in chains of retinaculum type with hairy surface (Figure 1A,B). The aerial mass color of the isolate S_10_Q_6_ was studied on different media. It was almost greenish grey in color on all the media used; therefore, it could be assigned to the grey color group. The color of substrate mycelium was almost yellow to brown on most of the media used (Table 1).

Concerning the physiological and biochemical activities of the isolate S_10_Q_6_, it showed positive results regarding the utilization of different carbon sources, coagulation, and peptonization of milk; catalase test; nitrate reduction; and hydrolysis of some polymers (casein, gelatin, cellulose, starch). However, it showed negative results with regard to H_2_S production; urease test; and utilization of L-arabinose, D-xylose, and rhamnose (Table 2). Temperature growth range was 25–35 °C. The analysis of cell wall composition indicated the presence of LL-diaminopimelic acid (DAP). The cultural, morphological and biochemical characteristics of the isolate S_10_Q_6_ indicated that this isolate belongs to Genus *Streptomyces.*

To complete the identification of the isolate S_10_Q_6_ at the species level, molecular identification by sequencing of 16S rRNA gene was used. DNA was extracted from exponentially growing culture of the S_10_Q_6_ isolate; Polymerase chain reaction (PCR) test was carried out for this target DNA using the primers given in Materials and Methods. The PCR products were electrophoresed using agarose gel (0.7%). DNA band of about 1445 bp (Appendix A) indicating a successful amplification of 16S rRNA gene was shown. This 16S rRNA gene was extracted from agarose gel, sequenced, and the gene sequence (Appendix A). was then submitted to Gene Bank under accession number KM192347. The cluster analysis was made using BLAST Programme, and the phylogenetic constructed tree is given in Figure 2. The pairwise similarity (99%) showed that the isolate S_10_Q_6_ belongs to *Streptomyces atrovirens* cluster. Consequently, the isolate S_10_Q_6_ was identified as one strain following *Streptomyces atrovirens* and designated *S. atrovirens* KM192347 (*S. atrovirens*).

### 2.3. Optimization of Anti-Dermatophytic Substance (s) Production by S. Atrovirens

Optimum antifungal activity obtained by *S. atrovirens* was achieved in starch nitrate broth adjusted initially at pH 7 and incubated at 30 °C in shaken cultures (150 rpm) for seven days (Table 2). The basal starch nitrate broth was used to evaluate the effect of different nutrients on antifungal substance production. Based on supplementation with and/or replacement of different nutrients, a medium designated modified starch nitrate medium (MSNM), was observed to maximize antifungal activity distinctively (*p* value ≤ 0.05) by *S. atrovirens*. The optimum composition of this medium is shown in Table 3. It contained starch (20 g/L) as the main carbon source, potassium nitrate (2 g/L) as the main nitrogen source, and dipotassium hydrogen phosphate (1 g/L) as main phosphorus source and NaCl at 0.5 g/L.

### 2.4. Extraction and Purification of the Antifungal Substance(s) from CFCS of S. Atrovirens

Solvent extraction of the antifungal substance(s) from CFCS obtained from the strain KM192347 was carried out as below described. The results given in Figure 3A show that both diethyl ether and ethyl acetate were the best two solvents for extraction of the antifungal substance(s) from CFCS of strain KM192347. Extractability of the tested solvents was arranged in following descending manner: diethyl ether > ethyl acetate > chloroform > benzene > petroleum ether > xylene > hexane > cyclohexane > toluene. As both diethyl ether and ethyl acetate showed the best extrability of the antidermatophytic substance produced by *S. atrovirens*, their extracts were combined, concentrated and eluted via column chromatography for further purification of the antidermatophytic substance(s). It was shown that the highest antifungal substance(s) were detected in the eluted fractions 14, 15, 16, and 17 (Figure 3B). The active fractions were combined, and the solvent of the eluent was evaporated at 40 °C in a vacuum oven. The product (white powder) of this process was used for latter experiments.

### 2.5. Elucidation of the Structure of the Antifungal Substance(s)

To test for existence of both nitrogen and sulfur, Lassaign’s test was used, where a green and purple color indicated the presence of nitrogen and sulfur. IR (KBr) discs (Figure 4A) gave bands at 3445 cm^−1^ (OH), 3158 cm^−1^ (NH), and 1616–1450 cm^−1^ (C=C, aromatic ring), in addition to bands at 1325 cm^−1^ (asym. SO_2_) and 1137 cm^−1^(-O-)ether. ^1^H NMR (DMSO-d_6_): δ = 2.48 (s, 1H, OH, exchange with D_2_O), 5.35 (s, 2H, *O*CH_2_, oxaborole ring), 5.26 (s, 1H, Ar-H_7_), 7.47 (d, 1H, *J* = 6.8 Hz, Ar-H_4_), 7.55 (d, 1H, *J* = 6.90 Hz, Ar-H_5_), 7.56–7.65 (m, 3H, Ar-H_3’,4’,5’_), 7.66–7.68 (m, 2H, Ar-H_2’,6’_) and 9.03 (s, 1H, NH, exchange with D_2_O) (Figure 4B). ^13^C NMR (DMSO-d_6_): δ = 68.5 (*O*CH_2_, oxaborole ring), 115.0, 119.8, 127.3, 127.6, 128.1, 129.2, 131.1, 131.8, 136.5, 139.7 (Ar-C, SP^2^ signals). Mass spectroscopy: M^+∙^ (m/e) = 289.06 (5.6%) as parent ion. Elemental analysis: Called C_13_H_12_ BNO_4_S (289.11): C, 54.01; H, 4.18; N, 4.84. Formed: C, 54.03; H, 4.15; N, 4.86 (Figure 4B).

For naming this antifungal compound, the preliminary formula obtained (C_13_H_12_ BNO_4_S) was constructed by Chem-Draw-Programme (Figure 5) andwas compared with the Novel Antibiotics Database (http://wwwO.nih.go.jp/~jun/NADB/search.html). It was shown that the antidermatophytic substance produced by *S. atrovirens*is [(N-(1-hydroxyl-1,3-dihydroxybenzo[c][1,2] oxaborole-6-yl) benzene sulphonamideis an oxaborole-6-benzene sulphonamide (OXBS)derivative.

### 2.6. Determination of the Minimum Inhibitory Concentration (MIC) Value of OXBS and Its Toxicity

MIC values were determined by tube dilution procedures (Table 3). It was showed that MIC of the substance OXBS produced by *S. atrovirens* is almost 20 μg/mL against the three dermatophytes tested.

The toxicity of OXBS was studied using white albino rats. No observed adverse effects were observed for a single dose (125, 250, and 500 μg/Kg wt) of OXBS throughout 14 days. In addition, no mortalities or hazardous signs in rats were observed throughout 14 days of observation after treatment of single dose of OXBS delivered by oral route. This clearly indicated that no potential toxicity of OXBS was observed for white albino rats. Histological examinations of liver tissues of white albino rats treated by OXBS are given in Figure 6. No differences were observed in the examined liver tissues between non-treated (control group) and treated by 500 μg/Kg wt.

## 3. Discussion

This study was an approach to investigate the potentiality of some locally isolated streptomycetes to produce other antidermatophytic substances that could contribute to saving humans and animals from infectious diseases. Three dermatophytic fungal strains were used as previously reported to cause many types of tinea [15]. In addition, *T. mentagrophytes* causes krion diseases (violent reaction results from its infection); the fungus can easily spread to other areas of the human body via socks, shoes, clothes, showers, bathtubs, or carpets [15]. In this regard, about 19.4% of the streptomycete isolates tested showed antidermatophytic activity. This is similar to the findings of a previous study [21] that found that 14% of streptomycetes tested inhibited some dermatophytes. However, previous study [5] reported that three out of 100 streptomycete isolates tested inhibited *T. mentagrophytes*. The percentage of inhibition of certain dermatophytes by streptomycetes differs among authors as it depends on the nature of both producer and indicator strains used. Hence, a comparison can be made when different authors use the same indicator dermatophytes. The antidermatophytic streptomycetes obtained, were isolated from soil and this concur with previous published results [22,23]. The isolate S_10_Q_6_was the most potent isolate and showed the higher antidermatopytic activity against the three indicator strains tested. It is of interest to find herein an inhibition of dermatophytes such as *T. mentagrophytes*, *M. canis*, and *T. tonsurans*, which causes many infections including many types of tinea and onychomycosis [15]. Consequently, the antidermatopytic strain S_10_Q_6_ was subjected to identification studies.

Considering the cultural, morphological and biochemical characteristics of the isolate S10Q6 and according to the criteria reported by previous studies [24,25,26,27] and based on the possible definitions of streptomycetes [28], the isolate S10Q6 could be classified as belonging to Genus: *Streptomyces.* In spite the use of recent molecular methods for identification of actinomycetes and other microorganisms, the cultural and morphological properties of the isolated organisms are still necessary to assign the color group of the isolated organism [29,30].

Previous studies have showed that the phenotypic identification could give elusive results [31]. Since phenotypic characteristics of the isolate S_10_Q_6_ were restricted to its identification at the Genus level, the molecular identification of the isolate S_10_Q_6_of the species level was mandatory and the molecular identification by sequencing of 16S rRNA gene approved that the strain S_10_Q_6_ belongs to *S. atrovirens*. This is because many species were transferred from their genera to other ones, and certain taxa were reassigned. For instance, Genus *Kitasatophora* was reassigned as sister taxon to Genus *Streptomyces* or a lineage that originated from *Streptomyces species* [32]. Similar recent studies also used these sequence of 16S rRNA gene as a successful way for the identification of actinomycetes [4,33].

Since culture conditions influence the production of antifungal substance(s) by streptomycetes, it was necessary to study the effect of such conditions on production of antidermatophytic substance(s) by *S. atrovirens*. Maximum antifungal substance was obtained in this study by *S. atrovirens* in MSNM adjusted initially at pH 7.0 and incubated by shaking at 150 rpm at 30 °C for seven days. These conditions appeared to be optimum for production of antifungals by streptomycetes [34]. The MSNM medium contained the optimum nutrients for production of maximum antifungal activity by *S. atrovirens*, and this is supported by previous results [35,36]. Moreover, the results of this study are confirmed by previous results that optimized C source [36], N source [37], phosphorous source [38], and mineral requirements [39].

Different results have stated that the highest antimicrobial activities were obtained using sucrose (1.0% *w/v*) as sole carbon source, peptone (1.0% *w/v*) as sole nitrogen source in cultures adjusted initially at pH 7.5 and incubated at 30 °C [40]. This might be due to the nature of the producer streptomycete used and different physiological conditions among streptomycete strains [34].

A two-step protocol including solvent extraction and column chromatography were used for purification of the antidermatophytic substance(s) from the CFCS of *S. atrovirens*. Due to existence of polar and non-polar chemical bonds and/or groups in many antibiotics, both polar and non-polar solvents were used for extraction throughout formation hydrogen bonds with the target substance [41]. Diethyl ether was showed to be the best solvent for the extraction of the antidermatophytic substance from CFCS of the strain *S. atrovirens*, which conforms with previously published results [42].

The reason for this is that the lone pair of electrons carried on the oxygen bridge of diethyl ether (C_2_H_5_-Ö_.._-C_2_H_5_) would make hydrogen bonds with positively charged chemical groups of the compound to be purified. Other solvents tested showed certain extrability to various degrees; this is possible because the extracted compound have both polar and non-polar moieties and tends to be of low polarity. Such properties enable the compound to make hydrogen bonds with these solvents, leading to extraction. In fact, the capability of solvents used for extraction from cell free culture filtrates of *Streptomycetes* spp. differs according to both solvents used and chemical nature of the antifungal agent produced [43]. These results support previous studies in this respect [44].

The aim of using both polar (ethanol) and non-polar (n-hexane) solvents (50:50 *v/v*) as eluent is to catch sharp fractions containing the bioactive compound(s) away from the impurities [45].

Elucidation of the chemical structure of unknown compounds needs the combination between several techniques. According to the data obtained from all instrumental analysis, the chemical formula of the antidermatophytic substance produced by *S. atrovirens*is C_13_H_12_ BNO_4_S; similar techniques were used by previous studies [20,46].

The obtained molecule was compared with others using the Novel Antibiotics Database. Based on this comparison, the antidermatophytic agent appeared to be an oxaborole-6-benzene sulphonamide derivative and named as N-(1- hydroxyl-1,3-dihydroxybenzo[c][1,2] oxaborole-6-yl) benzene sulphonamide. It was designated OXBS. Oxaborole- based chemical derivatives were discovered previously [47]; those authors found that the substance oxaborole-6-carboxamide could be used for treatment of trypanosomias is as an oral therapy. The previous study [48] discovered a potent benzoxaborole-based derivative with anti-pneumococcal activity in July 2014, and Anacor Pharmacyclics, Inc. [19] announced on the novel and first antidermatophytic fluorine-based oxaborole, which was designated tavaborole and was approved by the FDA in the same year [19]. It inhibits fungal peptide synthesis as it binds to the aminoacyl t- RNA synthetase. It is marked under the trade name ‘‘Kerydin” and is used to treat the fungal infections of the nail as well as nail bed that is termed as onychomycosis of the toe nail and solved the problem of inability of other antifungals that did not penetrate throughout nail bed, giving effective treatment of onychomycosis [15]

Tavaborole and other oxaborole-based derivatives are chemically synthesized, and it is promising to find out herein a biologically biosynthesized OXBS by *S. atrovirens*. The OXBS is different from tavaborole as it contains a benzene sulphonamide moiety at position 6 of the oxaborole ring and doesn’t contain a flourin atom at position five. Both tavaborole and OXBS contain the oxaborole-based moiety.

Sulphonamide existence in OXBS is very interesting as it showed greater antimicrobial activity [49]. Under normal temperature and pressure conditions, sulfanilamide is a stable substance [50].

MIC value of OXBS was 20 μg/mL. This is similar to the one obtained by previous study [2], but in involved about 25 μg/mL of cell free supernatants of two actinomycete isolates against the dermatophyte: *M.*
*canis* [5]. Travaborole exhibited MICs of ≤0.5 to 4 μg/mL against *M. canis* and *T.*
*tonsurans* and of 1–8 μg/mL against *T. rubrum* [15].

The comparison of different MIC values of different producer ctinomycete strains based on inhibitory activity against different sensitive organisms are highly speculative, as there are differences in the naturesof the sensitive strains used. Such comparison could only be done using the same compound and the same sensitive strains [51].

The results obtained herein show that OXBS has no toxicity on living tissues and this is similar to the non-toxic tavaborole [20]. This study reported the characterization of the antidermatophytic organism *S. atrovirens* KM192347. The antidermatophytic compound produced by *S. atrovirens* KM192347wasproduced maximally in MSNM and was characterized as OXBS with the chemical formula C_13_H_12_ BNO_4_S.

Further work will be necessary to study the effect and mode of action of OXBS on other indicator dermatophytes *in vitro* and *in vivo* as might be used in topical treatment of dermatophytes.

## 4. Materials and Methods

### 4.1. Isolation of Actinomycetes

Composite soil samples were collected from different cultivated lands of Sharkia Governorate; in particular 23 in Zagazig, 25 in Qenayat, 23 in Menya El-Kamh, and 24 in Belbis. All 95soil samples were processed and examined for possible actinomycetes isolation. Aliquots of about 1 Kg of composite soil samples were collected after removal and rejection of a two-inch layer from the soil. Each sample contained five mixed soil samples (200 g soil of each). Ten grams from each composite soil sample were suspended with 90 mL sterile distilled water. The suspensions were shaken for 30 min at 110 rpm and then serially diluted to 10^−6^. A 1mL sample from each dilution was transferred aseptically onto starch-nitrate agar plate and incubated at 30 °C for 7–21 days [52]. The incubation of culture under investigations was carried out in Digital Laboratory Incubators (New Brunswick Scientific Company, NJ, USA); incubators used were of different volumes and had microprocessors based control with auto-tune, digital temperature displays, and were double-walled with insulation provided with outer stainless steel metal door and inner glass viewing door.

During the incubation period, the appearance and growth of actinomycetes were observed daily. Colonies were recognized by their characteristic chalky to leathery appearance. Single colonies were picked up, and sub-cultured onto fresh starch nitrate agar medium and the purified cultures were transferred to slants of the same medium, incubated at 30 °C for 5–7days to achieve good sporulation, then refrigerated at 4 °C for the duration of the experiment.

### 4.2. Indicator Dermatophytes

The indicator dermatophytes used in this study (*T. tonsurans, M. canis*, and *T. mentagrophytes*) were provided from Microbiology Research Center, MIRCEN, (Culture Collection of Ain Shams University, Faculty of Agriculture, Ain Shams University, Cairo, Egypt).

They were stored in glass beads and subculturedonto soybean dextrose agar, which was used as dermatophytes selective agar [53,54]. The following medium components were weighed (g/L) and added to 1000 mL distilled water (papaic digest of soybean meal 10.0 g, dextrose 10.0 g, phenol red 0.2 g, agar-agar 20.0 g; all from Sigma). The pH value was adjusted to 7.0 ± 0.2, sterilized by autoclaving at 121 °C for 15 min, then cooled to 50 °C in digital water bath (New Brunswick Scientific Co. USA), and about 0.5 g cycloheximide was added aseptically.

### 4.3. Screening of Streptomycete Isolates for Production of Antifungal Substance(s)

Streptomycete isolates were screened for the production of antifungal metabolite(s) active against the dermatophytes used: *T. mentagrophytes*, *T. tonsurans*, and *M. canis*. Streptomycete isolates were inoculated as spore suspension onto starch nitrate agar media and incubated at 30 °C for 14 days. An agar cylinders (8mm) were cut out by a sterile metallic cork borer and transferred over dermatophyte selective medium (soybean dextrose agar) that were pre-seeded with the indicator dermatophytes [44,54]. Plates were kept at 4 °C for 4 h to allow the diffusion of fungal metabolites, then incubated at 28 °C for three weeks. Production of antifungal substance(s) was detected by the formation of inhibition zones around actinomycete agar discs. Isolate with the widest and highest activity was selected for further study concerning factors promoting its growth and antifungal production.

### 4.4. Cultural and Morphological Characteristics

For full description of the isolate S10Q6, its cultural characteristics such as color of colony (color of aerial mycelium), color of reverse side of colony (color of substrate mycelium) and color of diffusible pigments, have been studied according to previous study [52,55]. They were determined for 7–14 days old culture grown on the following media: tryptone yeast extract broth (ISP1) [56], yeast extract- malt extract agar (ISP2) [57], oat meal agar (ISP3) [58], inorganic salt starch agar (ISP4) [58], glycerol asparagine agar (ISP5) [59], peptone yeast extract iron agar (ISP6) [60], and tyrosine agar (ISP7) [61].The ability of the isolate S_10_Q_6_ to produce melanin pigment was tested according to previous study [52].

The spore chain morphology was evaluated by direct microscopic examination using 10 days old culture under a compound light microscope (Nikon, Tokyo, Japan) at 1000× magnification [52,55]. Spore surface was examined using electron microscope (Jeol, Tokyo, Japan) at 18,000× magnification by the spore-print technique [52]. Grids with collodion films were gently pressed onto the sporulating surface of oat meal agar cultures, shadowed with chromium and photographed.

### 4.5. Physiological and Biochemical Characteristics

The physiological and biochemical characteristics of the isolate S_10_Q_6_ were studied following the guidelines described previously [52,55,62]. The isolate S_10_Q_6_ was examined for the production of starch, cellulose, gelatin, casein hydrolysis, H_2_S, nitrate reduction, catalase production, urease test, peptonization and coagulation of milk, utilization of different carbon and nitrogen sources, temperature growth range, and cell wall hydrolysis for diaminopimelic acid.

### 4.6. Molecular Identification of the Isolate S_10_Q_6_

Molecular identification of the isolate S_10_Q_6_was determined using sequence of 16S rRNA gene. Total DNA was extracted from log phase culture of the isolate S10Q6. The 16S rRNA gene was amplified by PCR with universal primers (forward primer [F27] 5-AGAGTTTGATCCTGGCTCAG-3and reverse primer [R1492]5-GTTACCTTGTTACGACTT-3) as described previously [63]. The PCR products were electrophoresed via agarose gel and visualized under UV transillumination. The PCR products were purified using Gene JET™PCR Purification Kit (Fermentas). The amplified DNA fragments were partially sequenced by ABI 3730xl DNA sequence at GATC Biotech AG (Konstanz, Germany). The 16S rRNA gene sequence determined in the present study was recorded in Gen Bank with the accession number KM192347 on the NCBI web server (www.ncbi.nlm.nih.gov). Sequence analysis and its comparison to deposited data in Gene Bank were made using the Basic Local Alignment Search Tool (BLAST) Programme [64]. In addition, the sequence of 16S rRNA gene pattern of the KM192347strain was clustered with the one of *Streptomyces* spp. using Clusta 1X Programme. The dendrogram of the cluster analysis and the phylogenetic tree were constructed using Tree View X Phylogenetic Tree Programme.

### 4.7. Optimization of Both Environmental and Nutritional Conditions Necessary for Production of Antifungal Substance(s) by S. atrovirens

In this set of experiments, starch nitrate agar (Oxoid) was used as a basal medium. The agar well diffusion method was used to determine the optimum environmental and nutritional conditions. Aliquots of starch nitrate broth (250 mL for each) in 500 mL Erlenmeyer flasks (Gomhuria, Cairo, Egypt) were inoculated by (1% *v/v*) spore suspensions of actively growing cells in the mid-log phase after four days of growth of the *S. atrovirens* strain. The inoculated samples were incubated at different agitation rates (0, 50, 100, 150, 200, and 250 rpm) for different times (3–14 days), at different incubation temperatures (15, 20, 25, 30, 35, 40, 45, and 50 °C) and different pH values (4–12). Then CFCS were prepared by centrifuging the culture (5000Xg for 10 min at 4 °C). These CFCS were filter-sterilized using 0.45 μm Milipore filters (Amicon) and bioassyed immediately against the three dermatophyte strains used by the agar well diffusion assay as already reported [61]. The basal starch nitrate medium was used as a basal medium to optimize nutritional conditions (media components) at the optimum environmental conditions obtained. This basal medium was subjected to stepwise optimization with regard to different carbon sources, nitrogen sources, phosphorous sources, and different NaCl concentrations (0.1–5.0 g/L). Except for starch, other sugars were filter-sterilized using 0.45 µm (Milipore, Amicon). After incubation, CFCS were withdrawn, filter-sterilized, and assayed for their antifungal activity [65].

### 4.8. Extraction, Purification, and Structure Elucidation of the Antifungal Compound(s) Producedby S. atrovirens

Polar (ethyl acetate, chloroform, and diethyl ether) and non-polar organic solvents (benzene, petroleum ether, xylene, toluene, hexane, and cyclohexane) (Gomhoria, Cairo, Egypt) were tested for their efficiency to extract the active compounds from CFCS. An equal volume (100 mL each) of both CFCS and the individual solvent were mixed thoroughly by shaking in a separating funnel for 10 min, and after 15–30 min, the solvent layers were separated by rotary evaporator (Wheaton Eyela Rotary Vacuum Evaporator, NE-1-Tokyo Rikakikai Co. Ltd., Tokyo, Janpan) and kept for further study. The obtained extracts were combined and concentrated using rotary evaporator at 45 °C (50 rpm) under vacuum to the least volume (about 5 mL). For assaying extraction efficacy, filter paper discs (8 mm in diameter) were soaked in the obtained syrups, dried, and then placed onto a surface of agar plates that were freshly seeded with the tested dermatophytes. The diameter of the inhibition zones as indicated for extraction efficacy was measured after 4–6weeks of incubation at 28 °C. A control test for each solvent was also performed, as reported by previous study [66].

The crude antifungal substance OXBS was purified by column chromatography using silica gel (60–120 mesh) of column chromatography grade. A glass column (50 × 2.5 cm) was cleaned using water and rinsed with acetone. Silica gel was then packed in the column by using ethanol: n-hexane (50:50) as solvent system. The crude antifungal compound was loaded at the top of the column and eluted using the solvent system. Fractions were collected (3 mL). Antifungal activity against *T. mentagrophytes* was assayed for each fraction using agar well diffusion method. Active fractions were combined, and the solvent was evaporated at 40 °C in a vacuum oven. The product of this process was used for structure elucidation. NMR and MS were used to determine the compound skeleton. IR, which can identify the functional groups and carbon hybridization level, was phased out and mainly used as a compound characterization method, together with optical rotation; For structure elucidation of the purified antifungal compound, infra-red (IR), proton magnetic resonance (NMR), and mass spectrometry (MS) were carried out according to a previous study [67] in the Micro-Analytical Center, Cairo University, Egypt. The Chem-Draw Programme was used to construct the chemical formula of the chemical compound.

### 4.9. Determination of Minimum Inhibitory Concentration (MIC) Value of the Antifungal Substance

MIC values of OXBS were determined by tube dilution procedures according to CLSI (2008) [68]. The OXBS substance was serially diluted in DMSO: MeOH (Merck, Darmstadt, Germany) using sterile test tubes starting from the highest concentrations (100 μg/mL) to the lowest dilution (10 μg/mL). An inocula of the three indicator dermatophytes were tested of concentration about 10^4^CFU/mL [69] and then inoculated into tubes treated by the OXBS dilutions after incubation without agitation for three days at 25 °C. Growth of the indicator dermatophytes was then read visually without agitation with the aid of an inverted mirror [69,70]. The lowest concentration of the OXBS that inhibited growth of the three dermatophytes tested was recorded as the MIC. The antifungal agent Ketoconazole (10 μg/disc); DMSO: MeOH (Merck, Darmstadt, Germany) were used as a positive and negative controls respectively.

### 4.10. Toxcicity of Crude OXBS and Histological Examination

The Zagazig University Animal Care Board (Approval number: ZU-IACUC/1/F/82/2018) approved the design of the toxilogical animal experiment. The choice of animals used and design of the experiment were carried out according to a previous study [30]. Healthy male albino rats (*Rattus norvegicus*) were obtained from the Lab. Of Animal Health, Ministry of Health, Cairo, Egypt. Rats weighting 125 ± 10 g were housed in wire cages at room temperature (25 ± 5 °C) (five animals/cage). Animals were given food and water regularly and experienced 12 h light/dark cycles throughout the experimental period by oral route according to previous study in this respect [30,71,72,73]. Afteracclimatization, ratsweredividedintofourgroups (five rats for each). All treatments were delivered by gavages dissolved in 2 mL distilled water. The first group received 2 mL distilled water. Groups 2, 3, and 4 received one single dose of crude OXBS of about 125 µg/Kg, 250 µg/Kg, and 500 µg/Kg body weight, respectively. All rats were then kept under observation for 24 h to record any symptoms of toxicity or mortality and maintained for 14 days to observe behavioral and body weight changes.

For histological examination, liver tissue from rats of each group were taken immediately after slaughtering, washed with normal saline solution to remove blood, and then fixed in 10% neutral formalin and examined microscopically according to previous study [30].

### 4.11. Statistical Analysis

The collected data were tabulated and analyzed using IBM SPSS software. The results were expressed as a mean ± standard errors (SE) in either tables or figures. The Basic Local Alignment Search Tool Programme was used for cluster and pairwise analysis for identification of the streptomyceteorganism. The Chem-Draw Programme was used for the construction of the chemical formula of chemical compound(s).

### 4.12. Ethical Approval

Authors of this work throughout this study followed the policy of Zagazig University ZU-IACUC committee (Approval number: ZU-IACUC/1/F/82/2018) in the experiments carried out herein about experimental animals. A preliminary permission was taken from ZU-IACUC, which approved and reviewed this study to ensure that every animal is treated humanely and not subjected to unnecessary pain or distress and to use animal only if the study promises to contribute fundamental principles that are useful for more knowledge.

## 5. Conclusions

We considered *S. atrovirens* is as a source for production of antidermatophyticcompound. The purification and identification ofthe antifungal substance was carried out using several modern spectroscopic techniques such as IR (KBr), ^1^H NMR (DMSO-d_6_), ^13^C NMR (DMSO-d_6_), andmass spectroscopy. The pure substance was suggested to be C_13_H_12_ BNO_4_Sand designated asoxaborole-6-benzene sulphonamide (OXBS) derivative. OXBS is non-toxic to the liver of male Albino rats.

## Figures and Tables

**Figure 1 antibiotics-09-00176-f001:**
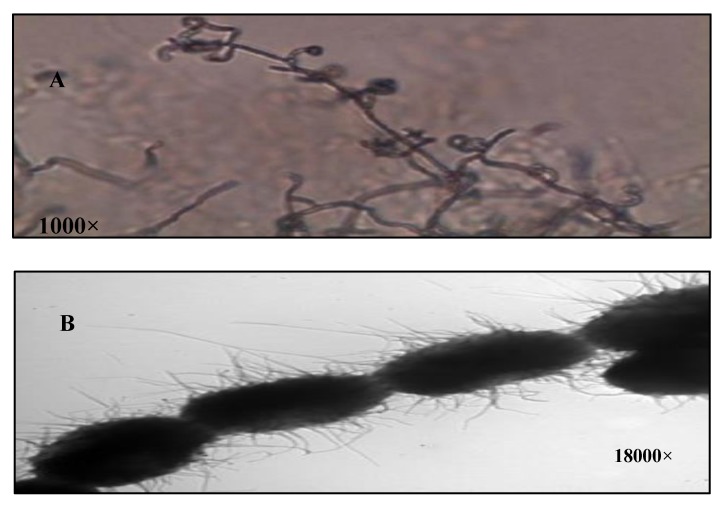
Characterization of *Streptomyces atrovirens* culture. (**A**): Spore chain morphology (1000×), (**B**): spore surface under electron microscope (18,000×).

**Figure 2 antibiotics-09-00176-f002:**
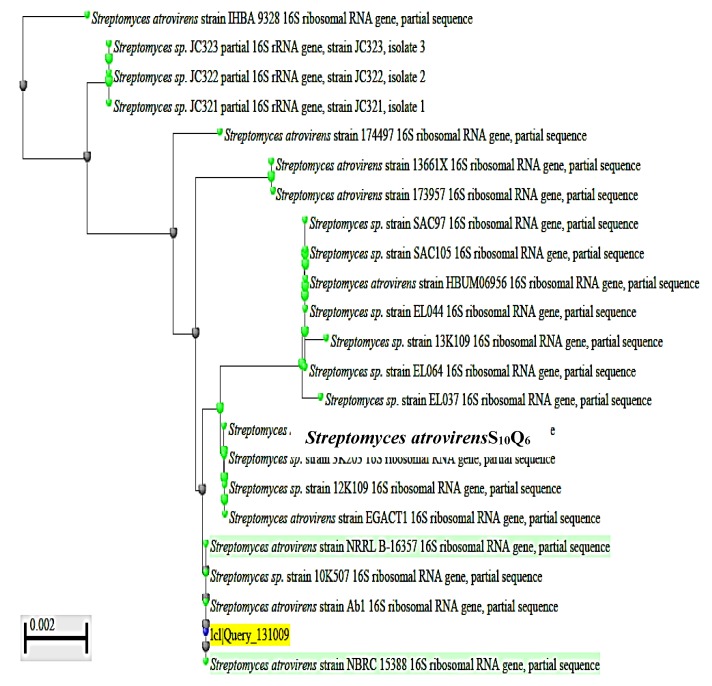
Phylogenetic tree showing the dendrogarm of the clusters analysis of some selected strains mentioned in the tree from the revelant taxa; the S_10_Q_6_ isolate existed within *S. atrovirens* cluster (99% similarity). The accession number KM192347 of NCBI web server (www.ncbi.nlm.nih.gov).

**Figure 3 antibiotics-09-00176-f003:**
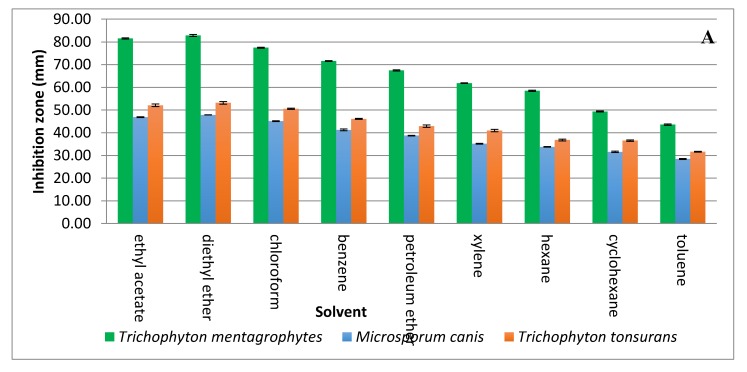
Extraction and purification of the antidermatophytic substance(s) produced by *S. atrovirens* (**A**); Antidermatophytic activity as showed by inhibition zone diameter of different solvent extracts of *S. atrovirens*; fractions eluted using column chromatography respectively (**B**). Results are the average of three replicates ± SE.

**Figure 4 antibiotics-09-00176-f004:**
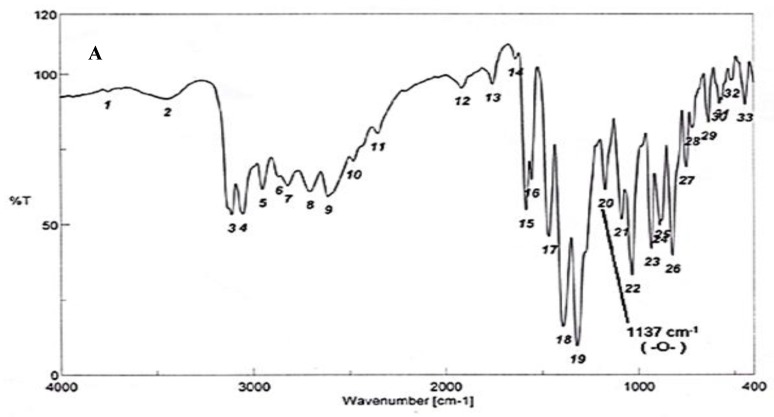
Elucidation of the structure of the antifungal substance(s). (**A**): Infrared (IR) spectrum of the antifungal compound produced by S.atrovirensKM192347. (**B**): ^1^H NMR spectrum of the antifungal compound produced by *S. atrovirens* KM192347 (DMSO-N_6_).

**Figure 5 antibiotics-09-00176-f005:**
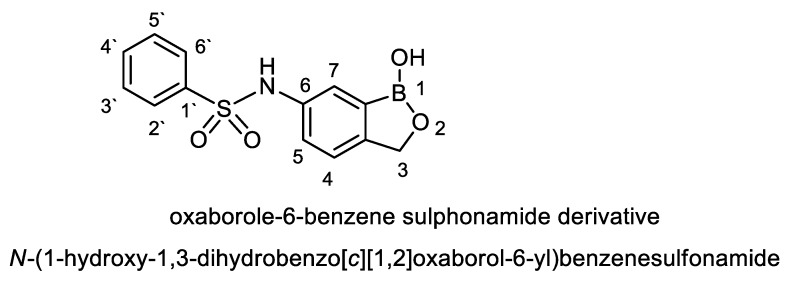
Structural formula of oxaborole-6-benzene sulphonamide (OXBS) derivative produced by *S. atrovirens*.

**Figure 6 antibiotics-09-00176-f006:**
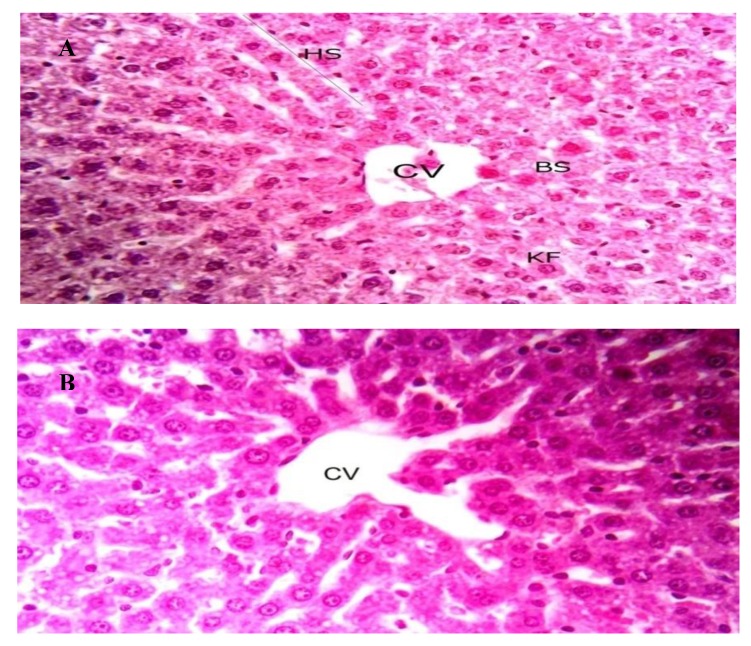
Liver sections from male Albino rats, X400. (**A**); control and (**B**) treated with OXBS by oral route. CV: central vein, HS: hepatic stand, BS: blood sinusoids, and KF: Kupffer cells.

**Table 1 antibiotics-09-00176-t001:** Actinomycte isolates with antidermatophytic activity against the tested dermatophyte strains (*T. mentagrophytes*, *M. canis* and *T. tonsurans*).

Area of Soil Sample	Serial Number and Isolate Code	Cultivated Plant	Color of Aerial Mycelium	Antifungal Activity (Inhibition Zone Diameter by mm)		
				*T. mentagrophytes*	*M. canis*	*T. tonsurans*
Zagazig (Z)	S_1_Z_3_	Onion	Grey	-	-	13.0 + 0.19
	S_3_Z_8_	Onion	Green	12.2 + 0.31	-	-
	S_3_Z_10_	Onion	Yellow	10.6 + 0.23	-	-
	S_5_Z_16_	Maize	Grey	-	-	10.4 + 0.33
	S_6_Z_19_	Maize	Yellow	15.0 + 0.87	-	19.2 + 0.41
	S_7_Z_23_	Maize	Green	-	-	11.0 + 0.34
Qenayat (Q)	S_10_Q_5_	Onion	Grey	13.8 + 0.45	-	-
	S_10_Q_6_	Onion	Grey	20.2 + 0.40	18.2 + 0.25	24.4 + 0.49
	S_12_Q_14_	Onion	Green	11.5 + 0.61	-	-
Menya EL-Kamh (M)	S_17_M_6_	Maize	Green	18.4 + 0.33	-	20.1 + 0.33
	S_19_M_11_	Maize	Green	-	-	12.8 + 0.30
	S_20_M_16_	Maize	Grey	13.9 + 0.43	12.9 + 0.52	17.6 + 0.62
	S_22_M_21_	Maize	Green	10.3 + 0.41	-	-
	S_22_M_26_	Maize	Green	21.0 + 0.39	-	21.9 + 0.45
	S_24_M_29_	Maize	White	11.2 + 0.53	-	-
Belbis (B)	S_26_B_4_	Maize	Grey	9.8 + 0.31	-	-
	S_26_B_6_	Maize	Yellow	8.9 + 0.63	-	-
	S_27_B_9_	Onion	Green	14.1 + 0.38	-	15.9 + 0.39
	S_27_B_10_	Onion	Grey	12.0 + 0.43	-	-
	S_28_B_17_	Maize	Green	-	-	11.4 + 0.23

Isolate show antifungal activity against the dermatophyte strains (-, no inhibition zone). Z: Zagazig City; 84Kmnorth Cairo. Q: Qenayat City; 85 Km north Cairo. M: Menya El-Kamh; 78 Km north Cairo. B: Belbis City; 68 Km north Cairo.

**Table 2 antibiotics-09-00176-t002:** The selection of optimum conditions necessary for anti-dermatophytic substance production by *S. atrovirens*.

Parameter	Optimum Value	Inhibition Zone * (mm)		
		*T. mentagrophytes*	*M. canis*	*T. tonsurans*
**Environmental Conditions**				
Incubation Period (Days)	7	40.0a ± 0.58	24.03b ± 0.26	26.17b ± 0.44
pH	7	39.27a ± 0.43	23.33b ± 0.6	24.87b ± 0.3
Incubation Temperature (°C)	30	40.03b ± 0.27	23.70c ± 0.75	26.23b ± 0.38
Agitation Rate (rpm)	150	55.23a ± 0.42	22.5c ± 0.36	26.77b ± 0.38
**Nutritional conditions**				
Carbon Source	Starch(20g/L)	52.80a ± 0.57	31.47c ± 0.26	34.63b ± 0.45
Nitrogen Source	Potassium Nitrate (2g/L)	52.37a ± 0.24	31.53c ± 0.29	35.33b ± 0.2
Phosphorus Source	DipotassiumHydrogen Phosphate (1g/L)	55.00a ± 0.25	32.73c ± 0.15	36.70b ± 0.65
Sodium Chloride Concentration (g/L)	0.5 g/L	50.73a ± 0.13	29.90c ± 0.21	33.43b ± 0.3

* Each value is an average of three replicates ± standard error (SE). Mean in the same row having different letters are significantly different (*p* ≤ 0.01).

**Table 3 antibiotics-09-00176-t003:** Minimum inhibitory concentration (MIC) values of the antifungal compound produced by *S. atrovirens*.

Concentration μg/mL	Growth of the Indicator Dermatophytes		
	*T. mentagrophytes*	*M. canis*	*T. tonsurans*
100	**−**	**−**	**−**
80	**−**	**−**	**−**
60	**−**	**−**	**−**
40	**−**	**−**	**−**
20	**−**	**−**	**−**
10	**+**	**+**	**+**
Negative Control	**+**	**+**	**+**
Ketoconazole (10 μg/disc)	**−**	**−**	**−**

**−**: No growth. **+**: positive growth.

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
