# Peer review of "Identification and Testing of Antidermatophytic Oxaborole-6-Benzene Sulphonamide Derivative (OXBS) from Streptomyces atrovirens KM192347 Isolated from Soil"

_antibiotics, 2020, doi:10.3390/antibiotics9040176_

Round 1
Reviewer 1 Report
The manuscript has elevated interest since it decribes a new compound produced by a strain of Streptomyces from Egipt. In terms of medicical interest I find the data well presented and discussed. I therefore agree with the publication of the manuscript at its present form.
Author Response
|
Replies to reviewer 1# Round 1 |
|
Comments and Suggestions for Authors The manuscript has elevated interest since it describes a new compound produced by a strain of Streptomyces from Egypt. In terms of medical interest I find the data well presented and discussed. I therefore agree with the publication of the manuscript at its present form.
|
|
Thanks for the thoughtful comments of the first reviewer.
|
Reviewer 2 Report
The work describes production, purification and identification of an anti-dermatophytic substance by Streptomyces atrovirens. A screening of 103 strains was performed towards antidermatophytic activity. The most potent isolate was identified (morphological and physiological identification, 16S rRNA gene sequencing) as Streptomyces atrovirens.
The Autors also conducted a selection of culture conditions (pH, incubation temperature, agitation rate, carbon, nitrogen and phosphorus source, sodium chloride concentration) for the antifungal substance production by the chosen strain, which was then characterized as an oxaborole-6-benzene sulphonamide derivative and designated OXBS with the chemical formula C13H12 BNO4S. Additionally, MIC values ​​against the dermatophyte cultures tested were determined
Please refer to the comments and questions below:
- In the text (lines 110-117) physiological and biochemical features of the strain were described, referring wrongly to the table 2, containing Antidermatophytic activity of the most active actinomycete isolates against the dermatophytes tested strains. There is no table containing features described in the text.
- The authors state that they have carried out optimization of the environmental and nutritional conditions to maximize production of the antifungal compound by atrovirens KM:192347, providing results of this experiment in the Tab. S3. In my opinion, optimization means searching for functions extremum, which is usually done using statistical methods. In the reviewed work, the selection of conditions for anti-dermatophytic substance production was carried out rather than the determination of optimal conditions. It is also a drawback that no results other than the best were shown here (tab. S3).
- Line 147: as described below, not above.
- Line 173: The figure number was incorrectly quoted - instead of 5B it should be 4B
- Line 386: What was the concentration of test strain spores in diffusion tests (agar well diffusion assay)? The quoted method from the literaturÄ™ position no. 61 it concerns identification of actinomycetes, but not the determination of the biological activity of their metabolites.
- Line 445: There is an unnecessary double fullstop.
- Line 446: it would be better to give the next number to the methodology paragraph, because the current entry suggests that the statistics concerned only the assessment of toxicity of the tested compounds.
Author Response
|
Replies to reviewer 2# Round 1 |
|
Are the results clearly presented? Can be improved |
|
Thank you for the reviewer. The results are reviewed again and the arrangement of Tables and Figures in the text is justified. Also, Tables and Figures legends and/or foot notes are reviewed again thoroughly with corrections of phrases and abbriviations. |
|
Please refer to the comments and questions below: · In the text (lines 110-117) physiological and biochemical features of the strain were described, referring wrongly to the table 2, containing Antidermatophytic activity of the most active actinomycete isolates against the dermatophytes tested strains. There is no table containing features described in the text.
|
|
Thanks for the thoughtful comments. The results of physiological and biochemical features of the strain were described in Table (2). It is corrected in the manuscript. |
|
· The authors state that they have carried out optimization of the environmental and nutritional conditions to maximize production of the antifungal compound by S. atrovirens KM:192347, providing results of this experiment in the Tab. S3. In my opinion, optimization means searching for functions extremum, which is usually done using statistical methods. In the reviewed work, the selection of conditions for anti-dermatophytic substance production was carried out rather than the determination of optimal conditions. It is also a drawback that no results other than the best were shown here (tab. S3).
|
|
Thanks for the logic comment. The selection of conditions for anti-dermatophytic substance production was carried out by S. atrovirens. The results of this investigation are given in Table 2. Corrected in the manuscript. The arrangement of Tables are justified in the text. Of course your opinion is correct, that statistical methods are used currently for optimization of conditions. The results obtained for optimization were conducted step wisely and practically using supplementation with and /or replacement of different nutrients (step by step). It gives results better than the using statistical methods. The optimum value tha gave the best antifungal activity is only written in (Table 2). This is because the length of manuscript cant seek all other results. For example, we studied the effect of pH 4-9, but pH 7 only is given as it is the optimum.
|
|
· Line 147: as described below, not above. |
|
Thanks for the thoughtful comment. It is corrected in the manuscript. |
|
· Line 173: The figure number was incorrectly quoted - instead of 5B it should be 4B |
|
Thanks for the thoughtful comment. (Figure 4B). It is corrected in the manuscript. |
|
Line 386: What was the concentration of test strain spores in diffusion tests (agar well diffusion assay)? The quoted method from the literaturÄ™ position no. 61 it concerns identification of actinomycetes, but not the determination of the biological activity of their metabolites. |
|
Aliquots of starch nitrate broth (250 mL for each) in 500 mL Erlenmeyer flasks (Gomhuria Co., Egypt) were inoculated by (1%V/V) spore suspensions of actively growing cells in the mid-log phase after 4 days of growth of the S. atrovirens strain. It is corrected in the manuscript. |
|
· Line 445: There is an unnecessary double fullstop. |
|
Thanks for the thoughtful comment. It is corrected in the manuscript. microscopically according to previous study [30].
|
|
· Line 446: it would be better to give the next number to the methodology paragraph, because the current entry suggests that the statistics concerned only the assessment of toxicity of the tested compounds.
|
|
Thanks for the thoughtful comment. It is corrected in the manuscript.
4.11. Statistical analysis · |
Reviewer 3 Report
The authors of this manuscript describe a discovery of a novel anti-dermatophyte drug from Streptomyces atrovirens. Authors describe the approach of the discovery, isolation, identification, structural moiety, MIC against major pathogenic dermatophytes, classification, and test for any potential major toxicity over the period of two weeks. However, following points need to be looked into and addressed prior acceptance. General English corrections throughout the manuscript may help.
- Table 1 and Table S1 are same! Where is the referred data?
- What is the difference between Table 1 and Table 2 other than repeating? Is there any value of having Table 2?
- Table S3 is same as Table 3!
- Where is Figure 5B (line 173).
- Please mention route in the line 193 or in figure legend too (including in the methods).
- Please cite and explain why 125/250/500ug/kg BW was chose for toxicity studies.
English language looked into throughout the MS. For eg., Line 22, 39, 51, 64, 69, 72, etc....
Author Response
|
Replies to reviewer 3# Round 1 |
|
Are the results clearly presented? Must be improved |
|
Thanks for the thoughtful comments of the reviewer. The results are reviewed again and the arrangement of Tables and Figures in the text is justified. Also, Tables and Figures legends and/or foot notes are reviewed again thoroughly with corrections of phrases and abbriviations. |
|
Comments and Suggestions for Authors The authors of this manuscript describe a discovery of a novel anti-dermatophyte drug from Streptomyces atrovirens. Authors describe the approach of the discovery, isolation, identification, structural moiety, MIC against major pathogenic dermatophytes, classification, and test for any potential major toxicity over the period of two weeks. However, following points need to be looked into and addressed prior acceptance. General English corrections throughout the manuscript may help. 1. Table 1 and Table S1 are same! Where is the referred data?
|
|
Thanks for the thoughtful comments. We are Sorry. We justified the sequence of Tables again and no supplementary tables are provided following remarks of reviewers. It is corrected in the manuscript. |
|
2. What is the difference between Table 1 and Table 2 other than repeating? Is there any value of having Table 2?
|
|
Thanks for this logic comment. Table 2 is deleted as its data are obivious in Table 1. |
|
3. Table S3 is same as Table 3! |
|
Thanks for the thoughtful comment. You are right. We justified the sequence of Tables again. No supplementary tables are provided following reviewers remarks. It is corrected in the manuscript. |
|
4. Where is Figure 5B (line 173). |
|
Thanks for the thoughtful comment. (Line 173: The figure number was incorrectly quoted - instead of 5B it should be 4B. (Figure 4B). It is corrected in the manuscript. |
|
5. Please mention route in the line 193 or in figure legend too (including in the methods). |
|
It is corrected in the manuscript. In addition, no mortalities or hazardous signs in rats were observed throughout 14 days of observation after treatment of single dose of OXBS delivered by oral route. It is also, corrected in the legend of Figure 6. |
|
6. Please cite and explain why 125/250/500ug/kg BW was chose for toxicity studies. |
|
Thanks for the thoughtful comment. why 125/250/500ug/kg BW was chose for toxicity studies. according to previous study in this respect[30, 71,72,73]. It is corrected in the manuscript.
|
|
7. English language looked into throughout the MS. For eg., Line 22, 39, 51, 64, 69, 72, etc.... |
|
Thanks for the thoughtful comment. English language of the manuscript was reviewed again. Line 22: Abstract: There is a need to continue research to find out other anti-dermatophytic agents to inhibit
Line 39: crude OXBS didn't show any toxicity on living cells. Finally the results obtained herein described Line 72: drugs. Due to such challenges and emergence of resistant variants of microorganisms [17], there is a |
Round 2
Reviewer 3 Report
English: Please rephrase.
Line 22
Line 39
Line 64
Line 72